# Effectiveness of percutaneous key lesion endoscopic lumbar decompression for the treatment of lumbar spinal stenosis in octogenarian patients

Chien-Tung Yang[1,2], Cheng-Che Hung[2,3], Chih-Ying Wu[4,5], You-Pen Chiu[1,2,6], Jeng-Hung Guo[1,2,6], Hui-Ru Ji[1,2,6], Cheng-Di Chiu[1,2,7,6,8]*

1 Department of Neurosurgery, China Medical University Hospital, Taichung, Taiwan, 2 Spine Center, China Medical University Hospital, Taichung, Taiwan, 3 Department of Neurosurgery, China Medical University Beigang Hospital, Chiayi, Taiwan, 4 Department of Neurosurgery, China Medical University Hsinchu Hospital, Hsinchu, Taiwan, 5 Graduate Institute of Integrated Medicine, China Medical University, Taichung, Taiwan, 6 Graduate Institute of Biomedical Science, China Medical University, Taichung, Taiwan, 7 School of Medicine, China Medical University, Taichung, Taiwan, 8 Graduate Institute of Medical Sciences, National Defense Medical Center, Taipei, Taiwan

* cdchiu4046@gmail.com

**Data Availability Statement:** For spinal MR and X-ray images used for measuring image parameters,

## Abstract

### Introduction

With increasing life expectancy, degenerative lumbar spinal stenosis (LSS) has become a common problem in the geriatric population. LSS reduces the quality of life, limits daily activities, and requires therapeutic aids. We share our experiences of treating octogenarian patients with LSS with key lesion percutaneous single portal endoscopic unilateral laminotomy and bilateral decompression (sEndo-ULBD).

### Materials and methods

Nine octogenarian patients who underwent sEndo-ULBD between January 2021 and July 2022 were prospectively enrolled in this study. Their visual analogue score (VAS), Oswestry Disability Index (ODI), disc height, spondylolisthesis, lumbar lordotic angle, lumbar scoliotic angle, and spinal canal area before and after sEndo-ULBD were followed up for more than six months.

### Results

The VAS score was significantly reduced three months after the operation (p < 0.05). The postoperative ODI scores of all patients improved relative to their preoperative scores; this difference became significant in the third month after the operation (p < 0.05). Index-level disc height did not significantly change after the operation. Spondylolisthesis, lumbar lordotic angle, and lumbar scoliotic angle showed no significant curve progression. The spinal canal area increased markedly after sEndo-ULBD (p <0.05), with no known surgery-related complications.

image files are available from figshare (DOI: 10. 6084/m9.figshare.25051325).

**Funding:** CDC was supported by China Medical University Hospital [DMR-110-217] for providing technical and financial support https://www.cmuh. cmu.edu.tw/Home/CmuhIndex_EN. The funders had no role in study design, data collection and analysis, decision to publish, or preparation of the manuscript.

**Competing interests:** The authors have declared that no competing interests exist.

## Conclusions

Key lesion sEndo-ULBD was an appropriate, safe, and effective treatment for octogenarian patients suffering from degenerative LSS. With an average follow-up of over one year, we did not find any significant progression in spinal curvature or instability. sEndo-ULBD is an ideal alternative to aggressive fusion fixation lumbar surgery for managing degenerative LSS in octogenarian patients with functional disability.

## Introduction

Increasing longevity is a global phenomenon facilitated by advancements in medical technology and socioeconomic progress. According to the World Health Organization, life expectancy rose significantly, by over six years, from 66.8 years to 73.4 years, between 2000 and 2019 [1]. In Taiwan, life expectancy had reached 80.86 years by 2021 [2]. However, population aging is accompanied by a rise in the number of frail elderly people and this increase is positively correlated with medical costs [3, 4]. The escalating healthcare expenses associated with the elderly in aging societies are a significant socioeconomic burden. In response to this concern, the Japanese Orthopedic Association introduced the concept of locomotive syndrome in 2007. Locomotive syndrome refers to a condition in which musculoskeletal dysfunction increases the likelihood of requiring nursing care. It is characterized by a decline in mobility-related functions, such as sit-to-stand movements and gait, resulting from dysfunctions in the musculoskeletal system, including bones, muscles, joints, and intervertebral discs [5]. It is crucial for the elderly to regain the ability to perform their daily activities and achieve self-care in order to reduce the burden on their families and the public care system.

Degenerative lumbar spinal stenosis (LSS) has a negative impact on quality of life (QoL) and is a critical risk factor of locomotive syndrome in elderly patients [6, 7]. The severity of LSS may be associated with the progression of locomotive syndrome [7]. Ideal lumbar spinal surgery for the elderly with LSS should alleviate locomotive syndrome and enable patients to regain physical function [8, 9]. However, most cases of degenerative LSS in the elderly involve multiple segments and osteoporosis, and are often associated with degenerative spondylolisthesis, degenerative kyphosis, or scoliosis, making the surgical operation more challenging [10]. When surgical intervention is necessary due to pain, neurological deficits, or severe disability, the choice of surgical method is crucial, especially for elderly patients. In addition, the debate whether the intervention for LSS should be decompression only or decompression plus fusion is ongoing. Many studies support the idea that decompression only is not inferior to decompression with fusion [11–14]. Considering the difficulties of surgery, osteoporosis, lengthy operation time, excessive intraoperative blood loss, and long hospital stay, decompression only may be superior to decompression with fusion [15–17]. Therefore, decompression on its own is viewed as a preferred option for geriatric patients with LSS.

Percutaneous endoscopic lumbar decompression is increasingly being used as a minimally invasive surgical technique that allows surgeons to decompress spinal stenosis through small incisions, thereby reducing excessive soft tissue damage [18–20]. Endoscopic spinal techniques offer early recovery and fewer surgical risks, especially for elderly patients [21, 22]. In the present study, we share our experience of performing key lesion percutaneous single portal endoscopic unilateral laminotomy and bilateral decompression (sEndo-ULBD) in octogenarian patients with degenerative LSS.

## Materials and methods

### Inclusion/exclusion criteria

The inclusion criteria were as follows: (1) patients aged over 80 years; (2) clinical symptoms characterized by lumbar radicular symptoms, e.g., radicular pain, paresthesia, or neurological claudication; (3) concordance between the key lesion in lumbar magnetic resonance imaging (MRI) and clinical symptoms and a diagnosis of degenerative LSS; (4) unsatisfactory conservative treatment for more than six months; and (5) at least six months of postoperative follow-up.

The exclusion criteria were as follows: (1) back pain or radicular pain caused by neoplasm, infection, or trauma; and (2) patients with mental illness or incapable of cooperation.

### Patient information

This study was conducted with the approval of the ethics committee of the China Medical University Hospital (IRB: CMUH110-REC2-1113). Nine octogenarian patients who underwent single-level key lesion sEndo-ULBD, performed by the same surgeon at our institution between 31 July 2021 and July 2022, were prospectively recruited. Nine patients achieved a 6-month follow-up, while only six patients achieved a 1-year follow-up because three of them were lost to follow-up. Furthermore, four patients have a 18-month follow-up period. Written informed consent was obtained from every patient who had a clear comprehension of the study details. The patients were followed up in the outpatient department after discharge. The patients' characteristics are described in Table 1. In addition to age and sex, data on the American Society of Anesthesiologists (ASA) physical status classification system level, comorbidities, length of hospital stay, intraoperative blood loss, bone mass density of the lumbar spine, and the index level were collected for all patients. Preoperative and postoperative MRI and plain and dynamic lumbar films were obtained for every patient. Since the patients were over 80 years of age, both degenerative scoliosis and multi-segment stenosis could also be revealed in image studies, as shown, for example, in Fig 1. The key lesion, on which the operation was based, was decided via thorough history taking, physical examination, and the consistency between images and symptoms. Transforaminal epidural steroid injection was not administered to these elderly patients due to refusal or concern about the side effects of the steroid.

**Table 1. Characteristics of geriatric patients with lumbar spinal stenosis.**

| No. | Gender | Age (years) | Comorbidities | ASA | LOS (days) | Blood loss (ml) | Level | BMD(T) | FU (months) |
|---|---|---|---|---|---|---|---|---|---|
| 1 | M | 88 | Inguinal hernia s/p OP | 3 | 9 | 10 | L3/4 | -0.4 | 12 |
| 2 | M | 83 | HTN, DM | 3 | 3 | 15 | L5/S1 | -1.8 | 6 |
| 3 | F | 80 | CKD; Liver cirrhosis s/p liver transplantation | 3 | 6 | 5 | L5/S1 | -1.9 | 18 |
| 4 | F | 84 | DM, HTN | 3 | 4 | 15 | L4/5 | -2.7 | 18 |
| 5 | M | 87 | DM | 3 | 7 | 10 | L5/S1 | -3.1 | 18 |
| 6 | F | 92 | HTN | 3 | 7 | 20 | L3/4 | -3.2 | 6 |
| 7 | M | 84 | CAD, CKD stage III, Brain aneurysm s/p OP | 3 | 3 | 20 | L4/5 | -0.2 | 18 |
| 8 | M | 80 | HTN, CAD, arrhythmia | 3 | 6 | 20 | L2/3 | -1.8 | 12 |
| 9 | F | 87 | HTN, aortic stenosis | 3 | 4 | 5 | L4/5 | -4 | 6 |
| Average | | 85.00 ± 3.91 | – | 3 | 5.44 ± 2.07 | 13.33 ± 6.12 | – | −2.1 ± 1.27 | 12.67 ± 5.57 |

M: male; F: female; OP: operation; s/p: state post; HTN: hypertension; CKD: chronic kidney disease; DM: diabetes mellitus; CAD: coronary artery disease; ASA: American Society of Anesthesiologists physical status classification level; LOS: length of hospital stays; BMD (T): bone mass density T score; FU: follow-up period.

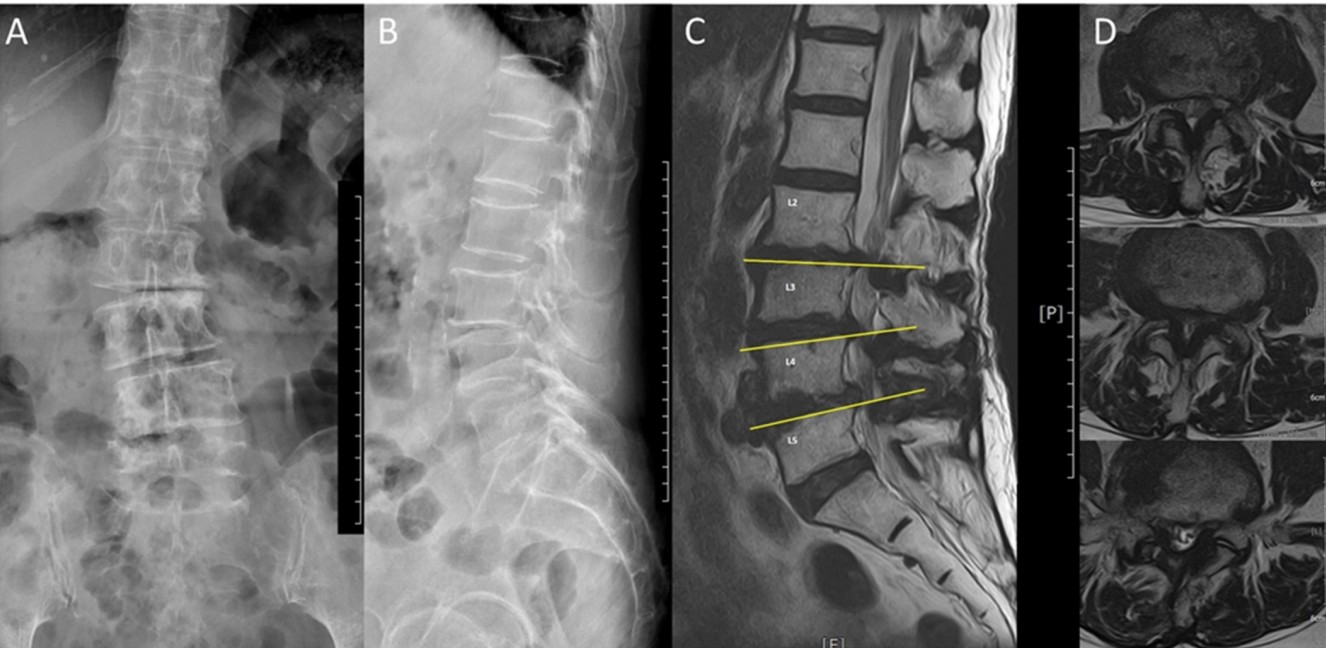

**Fig 1.** An example of degenerative scoliosis, spondylolisthesis, and associated severe lumbar spinal stenosis (A). The anteroposterior view of Lumbar spine X-ray shows degenerative scoliosis (B). The lateral view of Lumbar spine X-ray shows degenerative spondylolisthesis (C). The sagittal T2 Lumbar spine magnetic resonance image (MRI) shows multiple-level stenosis (D). The axial T2 Lumbar spine MRI shows the most severe stenosis located at the Lumbar 4/5 level.

## Surgical method

We used a 10-mm wide, 15° spine endoscope (SPINENDOS GmbH, München, Germany) and a cryoablation and radiofrequency ablation system (Ellman, New York, USA) for the procedure. We performed the key lesion sEndo-ULBD using a single portal endoscopic unilateral interlaminar approach for bilateral decompression (Fig 2). The procedure was performed under general anesthesia with the patient in the prone position on the operating table. We identified the index level of the interlaminar space with the assistance of fluoroscopy. A one-centimeter skin incision was made on the more severe symptom side and at the upper lateral edge of the interlaminar space. A unilateral approach to bilateral decompression was adopted while preserving bilateral facets. A discectomy was performed on a case-by-case basis depending on whether the bulging disc was causing nerve impingement.

## Clinical and image assessment

The visual analogue scale (VAS) and Oswestry Disability Index (ODI) were used to assess clinical outcomes. The VAS was used to evaluate the intensity of pain, which ranged from 0 to 10; the higher the score was, the greater the pain intensity. The ODI was employed to evaluate the level of functioning or disability during daily activities. The ODI questionnaire consists of 10 sections, each containing six statements with scores ranging from 0 to 5. The total score is calculated as total score/[5 × (total answered section)] × 100%. The levels of disability are in intervals of 20 points, with 0–20% being mild dysfunction, 21–40% being moderate dysfunction, 41–60% being severe dysfunction, and 61–80% being disability. Scores between 81–100% indicate that the patients are either bedbound or exaggerating their symptoms.

We were interested in the discrepancy or disparity in VAS and ODI before and after sEndo-ULBD. The image evaluation was categorized into single-index level and overall lumbar

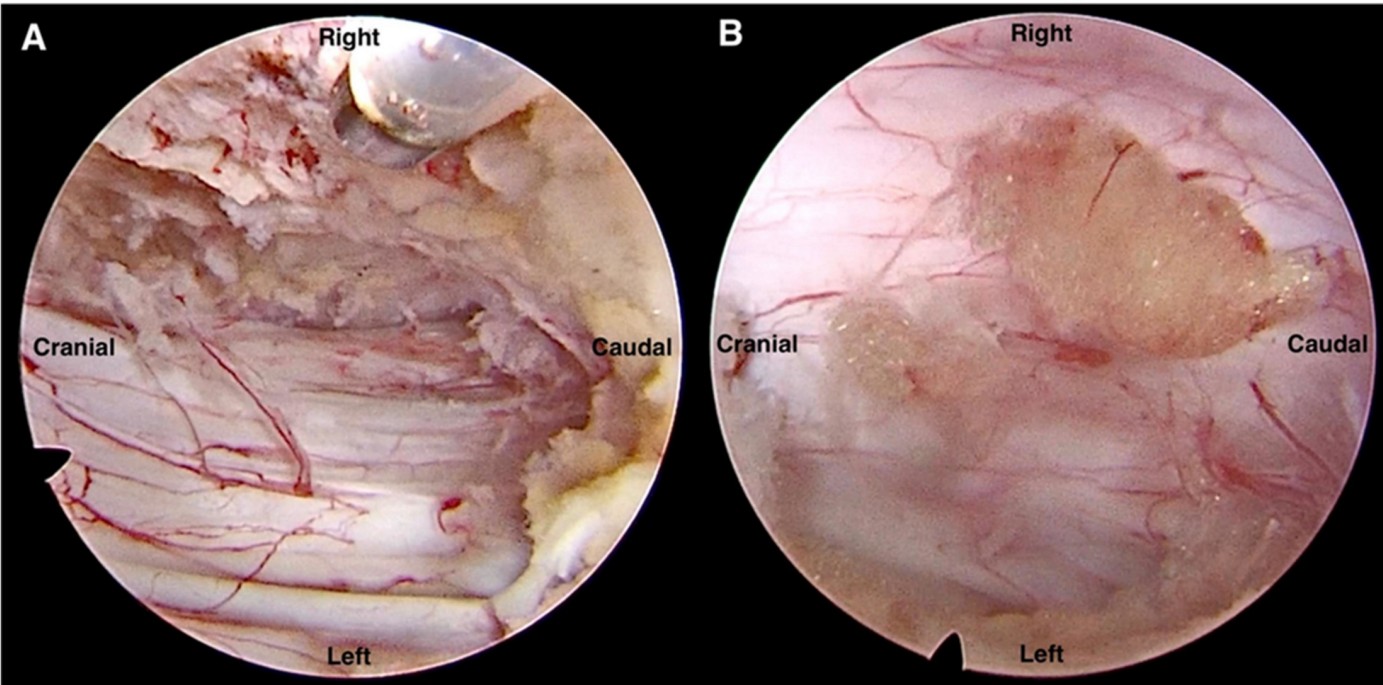

**Fig 2. Intraoperative image indicating bilateral decompression.** (A) Contralateral decompression and (B) ipsilateral decompression with discectomy.

curvature. Measurements were made with the built-in system in the Picture Archiving and Communication System (PACS, INFINITT PACS Ver. 3.011.4 BN2 64Bit). For single-index level assessment, we aimed at disc height, degree of spondylolisthesis, and spinal canal area. The index-level average disc height on the plain lumbar X-ray was measured. On the lumbar anteroposterior view, we measured bilateral disc height (A and C) and middle disc height (B) (Fig 3A). On the lateral view, anterior, middle, and posterior disc heights were measured (a, b, and c in Fig 3A). A total of six measurements were taken, and the average was defined as the disc height. Plain films of dynamic flexion and extension were also made to detect the degree of spondylolisthesis. The area of the spinal canal was evaluated using lumbar MRI before and after the operation. For the index level, we selected the most stenotic slice from the T2 axial MRI series taken prior to the operation. The change in the area of the spinal canal was then measured based on the relative position of the T2 axial MRI series six and eighteen months after the operation (Fig 3B).

The lumbar lordotic angle and scoliotic angle were obtained through overall lumbar curvature evaluation. The overall lumbar curvature was defined as the angle between the T12 lower endplate and S1 upper endplate on the lumbar plain radiograph. A line was drawn over the T12 lower endplate and another over the S1 upper endplate and the angle between these two lines was measured. The lumbar lordotic angle was measured on the lateral film for evaluation of kyphosis and the lumbar scoliosis angle was measured on the anteroposterior plain film for evaluation of scoliosis (Fig 3C).

## Statistical analysis

Data were statistically analyzed using IBM SPSS 20.0 software (International Business Machines Corporation, Armonk, New York). Paired-sample t-tests were used to compare data

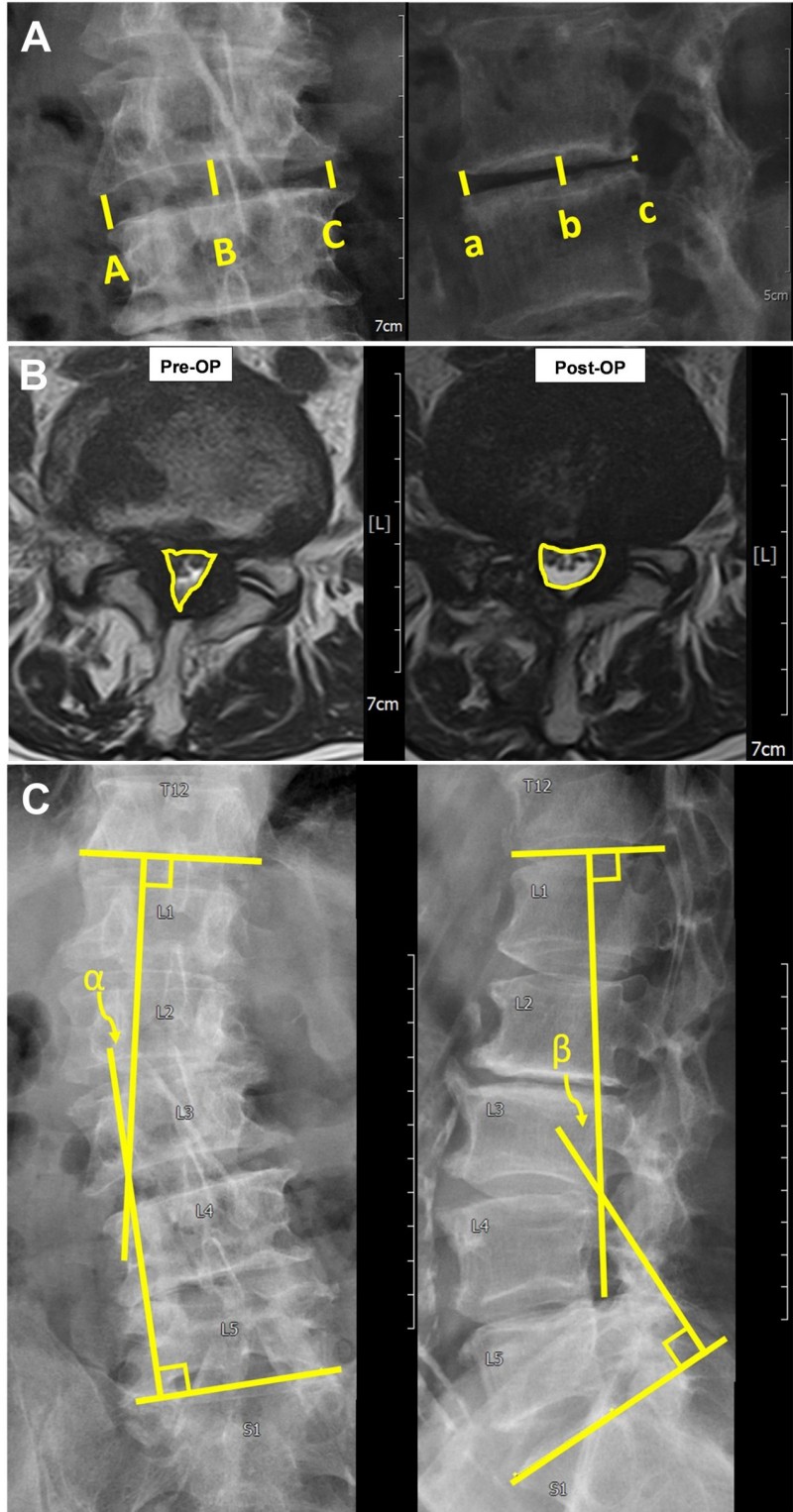

**Fig 3. A.** (Left) The anteroposterior view of the Lumbar spine X-ray. (Right) The lateral view of the Lumbar spine X-ray. Disc height is measured as the average length of A + B + C + a + b + c/6. **B.** The area of the index-level spinal canal (defined by the yellow line circling the spinal canal in lumbar T2 axial magnetic resonance image); the values are measured automatically by the built-in system in the Picture Archiving and Communication System (PACS) (INFINITT PACS Ver. 3.011.4 BN2 64Bit). **C.** Lumbar scoliotic and lordotic angle measurements. (Left) The lumbar

scoliotic angle between the lower endplate line of thoracic spine 12 and the upper endplate line of S1; the α angle is measured on the lumbar spine anteroposterior X-ray. (Right) The lumbar lordotic angle between the lower endplate line of thoracic spine12 and the upper endplate line of sacral spine 1; the β angle is measured on the lumbar spine lateral X-ray for evaluation of kyphosis.

before and after the operation for image assessment. We used one-way ANOVA for analyzing VAS and ODI. Dunn's multiple comparison test was used to analyze the differences between groups. Differences with two-tailed p-values of < 0.05 were considered statistically significant.

## Results

All patients successfully underwent the operation with satisfactory outcomes without any complications and were followed up for at least six months. The patients' average age was 85 ± 3.91 years (Table 1). All patients were classified as level III in the ASA physical status classification system, which means that they all had severe systemic disease.

Three single-index-level parameters were analyzed. The pre- and postoperative average disc heights did not differ significantly (p = 0.24, 0.08, and 0.20 at the 6-, 12-, and 18-month follow-ups, respectively; Table 2), indicating that disc height was not reduced after key lesion sEndo-ULBD. We further analyzed the sub-group that underwent discectomy; the disc height of this subgroup also did not show any further decrease after the procedure (p = 0.44, 0.28, and 0.25 at the 6-, 12-, and 18-month follow-ups, respectively). We compared the index-level area of the spinal canal (Table 2) before and after the operation; the area increased as much as by 73.67% and 88.58% at 6 months and 18 months, indicating significant improvement after effective decompression (p = 0.009 and p = 0.018, respectively). We did not observe any significant changes in spondylolisthesis on the dynamic plain film prior to or after the operation. Furthermore, no new cases of spondylolisthesis were detected (data not shown) at the index level during the postoperative follow-up period. Two overall lumbar curvatures were analyzed. The pre- and postoperative lumbar lordotic angles did not differ significantly (p = 0.91, 0.46, and 0.59 at the 6-, 12-, and 18-month follow-ups, respectively; Table 2). A similar result was obtained for the lumbar scoliotic angle (p = 0.15, 0.93, and 0.85 at the 6-, 12-, and 18-month follow-ups, respectively; Table 2).

The VAS and ODI were recorded before the operation, one day, and 1, 3, 6, 12, 18 months after the operation (Fig 4 and Table 3). The VAS score decreased significantly from 3 month to 18 months after the operation (p = 0.017, 0,002, 0.015, 6E-04, respectively). ODI scores also significantly improved from 3 month to 18 months after the operation (p = 0.014, 0.003, and 0,015, 0.016, respectively). The ODI and VAS scores suggest that function and pain levels of octogenarian patients with LSS improved after the key level sEndo-ULBD procedure.

## Discussion

The prevalence of degenerative LSS, a common health problem in the geriatric population, increases with age [23]. It is also the leading cause of lower back pain in elderly patients, resulting in neurological claudication and severe disabilities that negatively affect their QoL and often require medical or surgical aids [6–9]. Because medical costs of treating frail geriatric patients are high, surgical procedures that can help elderly patients to regain activities of daily living and self-care ability would not only benefit the QoL of this group but also reduce the economic burden. Although surgical treatment has been found to provide greater pain relief and functional improvement compared to conservative treatment in some studies [24, 25], a large cohort review suggested that surgical outcomes are not superior to those of conservative

**Table 2. Analysis of preoperative (Pre-OP) and postoperative (Post-OP) radiological parameters of geriatric patients with lumbar spinal stenosis undergoing key lesion percutaneous single portal endoscopic unilateral laminotomy and bilateral decompression.**

| No. | Disc height (mm) | | | | Area of the spinal canal (mm$^2$) | | | | |
|---|---|---|---|---|---|---|---|---|---|
| | Pre-OP | 6m | 12m | 18m | Pre-OP | 6m | 18m | Increased area (%) | |
| | | | | | | | | 6m/Pre-OP | 18m/Pre-OP |
| 1 | 8.84 | 7.95 | 8.15 | – | 64.03 | 96.4 | – | 50.55 | – |
| 2 | 7.7 | 8 | – | – | 83.86 | 102.3 | – | 21.99 | – |
| 3 | 5.65 | 5.95 | 5.66 | 5.51 | 104.16 | 198.56 | 194.22 | 90.63 | 86.46 |
| 4* | 7.73 | 5.99 | 6.33 | 6.20 | 58.4 | 95.54 | 93.99 | 63.60 | 60.94 |
| 5* | 9.26 | 9.94 | 9.24 | 9.17 | 54.87 | 94.97 | 98.15 | 73.08 | 78.88 |
| 6* | 7.61 | 7.46 | – | – | 40.66 | 118.3 | – | 190.95 | – |
| 7* | 7.71 | 7.44 | 7.31 | 7.25 | 48.1 | 106.88 | 109.69 | 122.20 | 128.05 |
| 8 | 10.09 | 9.49 | 9.74 | – | 83.84 | 95.67 | – | 14.11 | – |
| 9* | 8.62 | 8.43 | – | – | 114.08 | 155.06 | – | 35.92 | – |
| Average | 8.13 ± 1.27 | 7.85 ± 1.36 | 7.74 ± 1.61 | 7.03 ± 1.59 | 72.44 ± 25.44 | 118.19 ± 35.77 | 124.01 ± 42.27 | 73.67 ± 55.53 | 88.58 ± 44.61 |
| P | – | 0.24 | 0.08 | 0.20 | – | 0.009 | 0.018 | – | – |
| P* | – | 0.44 | 0.28 | 0.25 | – | – | – | – | – |

| No. | Lumbar lordotic angle | | | | Lumbar scoliotic angle | | | |
|---|---|---|---|---|---|---|---|---|
| | Pre-OP | 6m | 12m | 18m | Pre-OP | 6m | 12m | 18m |
| 1 | 47.36 | 49.68 | 48.97 | – | 0.48 | 1.12 | 0.88 | – |
| 2 | 37.49 | 44.82 | – | – | 5.49 | 6.23 | – | – |
| 3 | 22.19 | 32.21 | 27.41 | 25.70 | 1.81 | 2.56 | 2.51 | 2.78 |
| 4 | 43.27 | 38.34 | 32.92 | 43.71 | 5.71 | 10.81 | 5.11 | 4.76 |
| 5 | 50.93 | 55.94 | 47.83 | 47.77 | 8.25 | 7.55 | 7.49 | 8.01 |
| 6 | 61.06 | 58.13 | – | – | 3.82 | 3.58 | – | – |
| 7 | 44.5 | 35.46 | 39.28 | 38.92 | 9.09 | 9.4 | 10.01 | 9.65 |
| 8 | 50.25 | 47.78 | 51.26 | – | 2.58 | 2.35 | 2.09 | – |
| 9 | 22.79 | 19.66 | – | – | 10.09 | 11.93 | – | – |
| Average | 42.20 ± 12.89 | 42.45 ± 12.27 | 41.28 ± 9.67 | 39.03 ± 9.59 | 5.26 ± 3.38 | 6.17 ± 3.98 | 4.68 ± 3.53 | 6.30 ± 3.10 |
| P | | 0.91 | 0.46 | 0.59 | | 0.15 | 0.93 | 0.85 |

P value was compared with Pre-OP data.

* Patients who received discectomy

Increased area (%) = (Post-OP area)—(Pre-OP area)/(Pre-OP area) × 100

treatment and that the complication rate of surgical treatment for LSS is as high as 10–24% [26]. Spinal disorders in elderly people are also complicated by degenerative kyphoscoliosis and degenerative spondylolisthesis [27]. In addition, given the coexistence of complicated medical problems, slow recovery, and high surgical and anesthesia risks, most physicians prefer more conservative treatment to surgery for elderly patients with LSS. However, these choices may not be contradictory since minimally invasive spinal surgical procedures that can achieve satisfactory results without lengthening hospital stay and increasing intraoperative blood loss are increasingly becoming available [28].

To the best of our knowledge, we are the first to address single-level key lesion decompression via sEndo-ULBD in multi-segment LSS with or without degenerative deformities in octogenarians. Instead of long-/short-segment decompression plus fusion, decompression on its own has advantages in reducing both intraoperative blood loss and hospital stay [15–17]. Previous studies have also demonstrated the effectiveness of endoscopic lumbar decompression surgery [18, 19]. Key lesion sEndo-ULBD echoes the superiority of multiple-segment fixation,

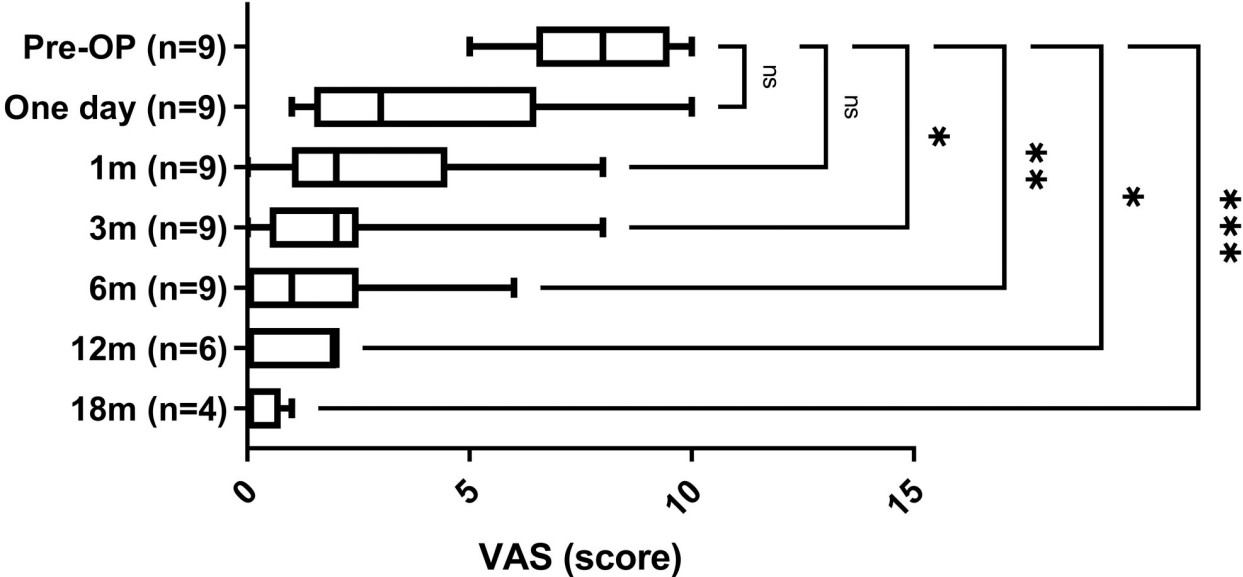

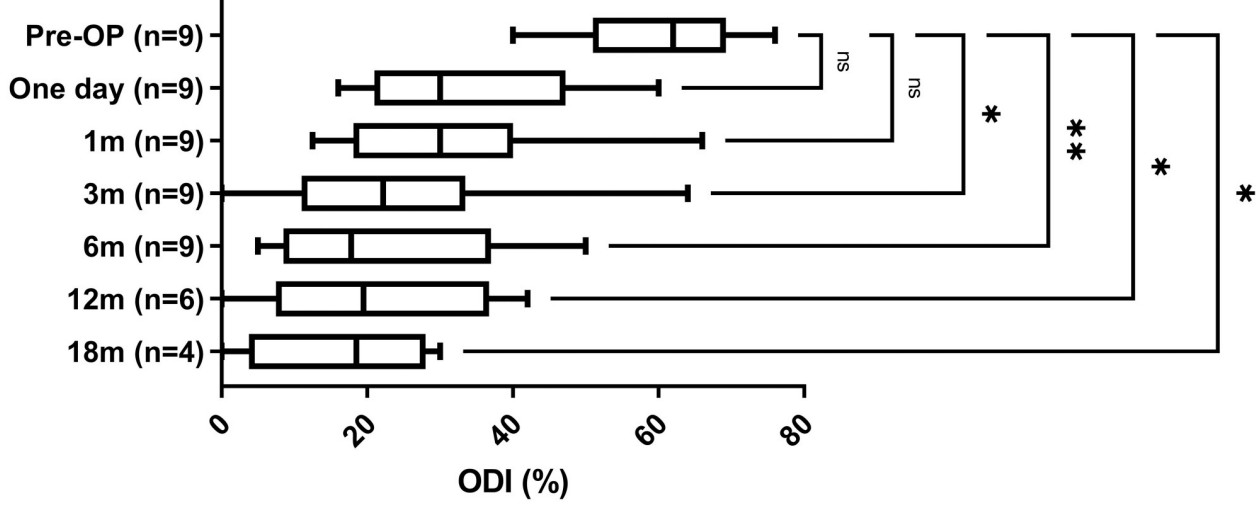

**Fig 4. The pre- and post-surgery visual analogue score (VAS) and Oswestry Disability Index (ODI).** *, **, and *** represent p < 0.05, 0.01, and 0.001 compared to pre-operation.

correction, and fusion surgery in lowering intraoperative blood loss (13.33 ml on average), shortening hospital stays (5.44 days on average), and faster recovery. We chose single-level key lesion causing symptoms to treat in order to decrease surgical time and surrounding tissue damage. Our data showed significant VAS and ODI relief three months after surgery, with both remaining at a steady satisfactory state for eighteen months.

Fusion is not the only choice for treating scoliosis and spondylolisthesis. Mori et al. suggested that minimally invasive decompression for degenerative spondylolisthesis does not

**Table 3. VAS and ODI value at Pre-OP, one day, 1 month, 3 months, and 6 months after operation.**

| No. | VAS | | | | | | | ODI (%) | | | | | | |
|-----|--------|---------|-----|-----|-----|-----|-----|--------|---------|-------|-------|-------|-----|-----|
| | Pre-OP | One day | 1m | 3m | 6m | 12m | 18m | Pre-OP | One day | 1m | 3m | 6m | 12m | 18m |
| 1 | 8 | 1 | 1 | 2 | 1 | 2 | – | 64 | 18 | 14 | 12 | 12 | 10 | – |
| 2 | 7 | 3 | 3 | 3 | 3 | – | – | 48 | 30 | 30 | 30 | 30 | – | – |
| 3 | 10 | 3 | 1 | 1 | 1 | 2 | 1 | 76 | 30 | 36 | 22 | 14 | 24 | 22 |
| 4 | 8 | 6 | 4 | 2 | 1 | 2 | 0 | 62 | 50 | 40 | 36 | 44 | 42 | 30 |
| 5 | 5 | 2 | 0 | 0 | 0 | 0 | 0 | 54 | 24 | 22.22 | 22.22 | 17.78 | 15 | 15 |
| 6 | 6 | 10 | 8 | 8 | 6 | – | – | 66 | 60 | 66 | 64 | 50 | – | – |
| 7 | 9 | 4 | 2 | 0 | 0 | 0 | 0 | 57.78 | 16 | 12.5 | 0 | 5 | 0 | 0 |
| 8 | 9 | 7 | 5 | 2 | 2 | 2 | – | 40 | 44.44 | 26.67 | 31 | 20 | 35 | – |
| 9 | 10 | 1 | 1 | 1 | 0 | – | – | 72.5 | 40 | 40 | 10 | 5 | – | – |

accelerate postoperative slippage compared to the natural course [29]. Ravinsky et al. also reported that there is no correlation between radiographic slip progression and symptomatic worsening after minimally invasive decompression without fusion for low-grade degenerative lumbar spondylolisthesis [30]. Furthermore, the progression of scoliosis curves after decompression surgery is similar to natural progression [31, 32]. Octogenarians have a limited life expectancy, and the progression of slippage, kyphosis, or scoliosis curves may not be a major concern. In our study, we did not observe any new progression of spondylolisthesis, kyphosis, or scoliosis within 1.5 years after sEndo-ULBD (Table 2). Also, up to the last documented follow-up in our patient series, none required additional surgery for a residual stenotic lesion. In addition, the change in disc height with or without discectomy was not statistically significant. Therefore, our findings imply that decompression on its own could potentially serve as a viable alternative to fusion fixation surgery. Key lesion sEndo-ULBD, as minimally invasive surgery, is a good option for geriatric patients, because it reduces the risks of fusion surgery, such as higher blood loss and longer operative and hospital stay times. Furthermore, it can lead to improvements in the QoL of octogenarians suffering from LSS.

However, our study has some limitations. First, the sample size was small. Second, the follow-up time should have been longer. Third, the frequency of follow-up should have been higher since there could have been further events that negatively affected the octogenarians. In our series, patient-6 suffered from a fall that resulted in a T12 compression fracture about one month after the operation, this insult might have affected the result. Fourth, this was a prospective study, rather than a randomized controlled study, which could have biases in some areas, such as patient selection and interpretation of results. Fifth, we must acknowledge the limitation of not specifying how many patients with multi-segment stenosis agreed or disagreed with undergoing key lesion decompression in this study.

## Conclusion

Key lesion sEndo-ULBD can improve the QoL of octogenarian patients suffering from degenerative LSS without exacerbating spondylolisthesis, kyphosis, or scoliosis. It is a good surgical option when conservative medical treatment has failed.

## Acknowledgments

We would like to express our gratitude to Chung-Yu Huang and Jia-Jie Huang for their valuable administrative and technical support.

## Author Contributions

**Conceptualization:** Cheng-Di Chiu.

**Data curation:** Hui-Ru Ji.

**Formal analysis:** You-Pen Chiu.

**Funding acquisition:** Cheng-Di Chiu.

**Methodology:** Chih-Ying Wu, Jeng-Hung Guo.

**Supervision:** Cheng-Che Hung.

**Writing – original draft:** Chien-Tung Yang.

**Writing – review & editing:** Chih-Ying Wu, Jeng-Hung Guo, Cheng-Di Chiu.

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
