## [Decision Letter · Decision Letter 0]

20 Dec 2023

PONE-D-23-21132Effectiveness of percutaneous key lesion endoscopic lumbar decompression for the treatment of lumbar spinal stenosis in octogenarian patientsPLOS ONE

Dear Dr.Chiu,

Thank you for submitting your manuscript to PLOS ONE. After careful consideration, we feel that it has merit but does not fully meet PLOS ONE’s publication criteria as it currently stands. Therefore, we invite you to submit a revised version of the manuscript that addresses the points raised during the review process.

We look forward to receiving your revised manuscript.

Kind regards,

Tadashi Ito

Academic Editor

PLOS ONE

Journal Requirements:

"This study is supported by funding from China Medical University Hospital (DMR-110-217). We would like to express our gratitude to Chung-Yu Huang and Jia-Jie Huang for their valuable administrative and technical support."

"CDC was supported by China Medical University Hospital [DMR-110-217] for providing

technical and financial support https://www.cmuh.cmu.edu.tw/Home/CmuhIndex_EN.

The funders had no role in study design, data collection and analysis, decision to

publish, or preparation of the manuscript"

3. We note that your Data Availability Statement is currently as follows: [Yes - all data are fully available without restriction]

Reviewers' comments:

Reviewer's Responses to Questions

**Comments to the Author**

1. Is the manuscript technically sound, and do the data support the conclusions?

Reviewer #1: Partly

Reviewer #2: Yes

Reviewer #3: Yes

2. Has the statistical analysis been performed appropriately and rigorously? 

Reviewer #1: Yes

Reviewer #2: Yes

Reviewer #3: Yes

3. Have the authors made all data underlying the findings in their manuscript fully available?

Reviewer #1: Yes

Reviewer #2: Yes

Reviewer #3: Yes

4. Is the manuscript presented in an intelligible fashion and written in standard English?

Reviewer #1: No

Reviewer #2: No

Reviewer #3: Yes

5. Review Comments to the Author

Reviewer #1: This study was designed to clarify the efficacy of the key lesion decompression for lumbar spinal stenosis at multiple segments in image studies using a minimally invasive percutaneous endoscopic technique (sEndo-ULBD). The key lesion was decided by history taking, physical examination and the consistency between symptoms and images. After the authors evaluated the radiographic and patient-reported clinical outcomes pre- and post-operatively in nine older patients, they concluded that key lesion sEndo-ULBD was appropriate, safe and effective treatment for degenerative LSS.

Major comment:

The authors should be commended for their efforts on this article, which required substantial effort. However, I am concerned regarding the following issues.

1. The reviewer also think that key lesion decompression using a minimally invasive surgical technique is true minimally invasive treatment. The main purpose of the current study might be to clarify the efficacy of the key lesion decompression for lumbar spinal stenosis at multiple segments in image studies. If so, a large number of patients with long-term follow-up are needed. Remaining stenotic lesions may cause symptoms 1, 2 or 5 years after index surgery even in octogenarian patients. 6-month follow-up is too short and insufficient. Did any patients, who is included or not included in this series, need to undergo additional surgery for remaining stenotic lesions during longer follow-up period?

2. Key lesion decompression may be sometimes challenging. Some older patients with lumbar spinal stenosis at multiple segments may have difficulty in choosing key lesion (selective) decompression to avoid additional surgery in the future after index surgery. How many patients with multi-segment stenosis agreed or disagreed with undergoing key lesion decompression?

Specific comment:

1. Page 2, Line 53 (in the Conclusion section of Abstract)

The authors described that the long-term follow-up did not find any significant progression in spinal curvature or instability. The reviewer think that the term “long-term” is not appropriate. 6-month should be short-term.

Reviewer #2: This paper was about”Effectiveness of percutaneous key lesion endoscopic lumbar decompression for the treatment of lumbar spinal stenosis in octogenarian patients”

Now degenerative lumbar spinal stenosis (LSS) was a big problem. It affects the quality of life and limits daily activities. The sEndo-ULBD is very suitable for treating octogenarian patients with LSS .

Key lesion sEndo-ULBD was an appropriate, safe, and effective treatment for octogenarian patients suffering from degenerative LSS. sEndo-ULBD is a good choice to to aggressive fusion fixation lumbar surgery for managing degenerative LSS in octogenarian patients with functional disability.

Limit: The number of patients was too little and follow-up need more long time.

Reviewer #3: THE MANUSCRIPT EXPLAINS IN DETAIL THE NEED FOR THE STUDY , THE BENEFITS OF THE STUDY , THE OUTCOME OF THE PROCEDURES AND NEED FOR PARADIGM SHIFT IN THE MANAGEMENT OF THESE CONDITIONS IN THE ELDERLY.IT IS THUS WORTH PUBLISHING

6. PLOS authors have the option to publish the peer review history of their article (what does this mean?). If published, this will include your full peer review and any attached files.

Reviewer #1: No

Reviewer #2: No

Reviewer #3: **Yes: **MUSTAPHA ALIMI

---

## [Author Response · Author response to Decision Letter 0]

3 Feb 2024

Reviewer #1:

 This study was designed to clarify the efficacy of the key lesion decompression for lumbar spinal stenosis at multiple segments in image studies using a minimally invasive percutaneous endoscopic technique (sEndo-ULBD). The key lesion was decided by history taking, physical examination and the consistency between symptoms and images. After the authors evaluated the radiographic and patient-reported clinical outcomes pre- and post-operatively in nine older patients, they concluded that key lesion sEndo-ULBD was appropriate, safe and effective treatment for degenerative LSS.

Major comment:

The authors should be commended for their efforts on this article, which required substantial effort. However, I am concerned regarding the following issues.

1. The reviewer also think that key lesion decompression using a minimally invasive surgical technique is true minimally invasive treatment. The main purpose of the current study might be to clarify the efficacy of the key lesion decompression for lumbar spinal stenosis at multiple segments in image studies. If so, a large number of patients with long-term follow-up are needed. Remaining stenotic lesions may cause symptoms 1, 2 or 5 years after index surgery even in octogenarian patients. 6-month follow-up is too short and insufficient. Did any patients, who is included or not included in this series, need to undergo additional surgery for remaining stenotic lesions during longer follow-up period?

Response: 

We appreciate the reviewer's valuable comments. 

We acknowledge that a 6-month follow-up may not be sufficient for a more comprehensive analysis. Our data collection commenced from January 2021 to July 2022. The submission period of our work allowed us to extend the follow-up duration for our patients. We continued collecting data until December 2023, enabling us to include outcomes for 6 patients who were followed up for 1 year and 4 patients for 1.5 years. (Table 1) However, the total number of patients did not increase, as we maintained a minimum follow-up period of 6 months. In the case of octogenarian patients, regular follow-up posed challenges, likely due to their complex medical issues and the potential need for assistance from younger companions for hospital visits.

As far as our records show, up to the last documented follow-up for our patient series, none required additional surgery for residual stenotic lesions. We add this statement in the “Discussion” paragraph (Lines 281-282)

2. Key lesion decompression may be sometimes challenging. Some older patients with lumbar spinal stenosis at multiple segments may have difficulty in choosing key lesion (selective) decompression to avoid additional surgery in the future after index surgery. How many patients with multi-segment stenosis agreed or disagreed with undergoing key lesion decompression?

Response: 

We thank the reviewer for their significant remarks. 

We acknowledge the shortfall in providing an exact count of all patients and add this statement to the part of the study limitation. (Lines 294-296) Traditionally, in Taiwan, surgery for octogenarian patients is often not considered a viable option. This perception stems from both patients and their families, who commonly view spine surgery as excessively risky and, culturally, as an omen of bad luck. Consequently, these octogenarian patients are more inclined to seek less invasive alternatives, such as traditional Chinese medicine or manipulation therapies.

Furthermore, long-segment spinal stenosis surgery presents even higher risks due to extended surgical times and increased blood loss, compared to key lesion endoscopic surgery. Based on our experience, it is rare for either octogenarian patients or their families to consent to long-segment spinal surgery. These factors collectively contribute to the limited number of cases in our series.

Specific comment:

1. Page 2, Line 53 (in the Conclusion section of Abstract)

The authors described that the long-term follow-up did not find any significant progression in spinal curvature or instability. The reviewer thinks that the term “long-term” is not appropriate. 6-month should be short-term.

Response: 

Thank you for the reviewer's kind reminder. 

We have addressed the previously misleading term and revised "long-term" to "With an average of one year’s follow-up" (Line 53-54).

Reviewer #2: 

This paper was about”Effectiveness of percutaneous key lesion endoscopic lumbar decompression for the treatment of lumbar spinal stenosis in octogenarian patients”

Now degenerative lumbar spinal stenosis (LSS) was a big problem. It affects the quality of life and limits daily activities. The sEndo-ULBD is very suitable for treating octogenarian patients with LSS .

Key lesion sEndo-ULBD was an appropriate, safe, and effective treatment for octogenarian patients suffering from degenerative LSS. sEndo-ULBD is a good choice to to aggressive fusion fixation lumbar surgery for managing degenerative LSS in octogenarian patients with functional disability.

Limit: The number of patients was too little and follow-up need more long time.

Response:

Thank you for the reviewer's advice. 

We recognize that a 6-month follow-up may not adequately suffice for a more comprehensive analysis. Our data collection spanned from January 2021 to July 2022. The submission period of our work provided an opportunity to prolong the follow-up duration for our patients. We continued to gather data until December 2023, which allowed us to include outcomes for 6 patients with a 1-year follow-up and 4 patients with a 1.5-year follow-up. (Table 1) Nevertheless, the overall patient count remained unchanged, as we adhered to a minimum follow-up period of 6 months. Regular follow-ups for octogenarian patients were challenging, often due to their intricate medical conditions and the likely necessity for younger companions to assist with hospital visits.

Our next step is to recruit more octogenarian patients with multilevel spinal stenosis and to collaborate with additional spine centers. This effort is aimed not only at increasing our patient cohort but also at extending the follow-up duration. This will help us gain a clearer understanding of the effectiveness and efficiency of key-lesion endoscopic surgery.

Reviewer #3:

 THE MANUSCRIPT EXPLAINS IN DETAIL THE NEED FOR THE STUDY , THE BENEFITS OF THE STUDY , THE OUTCOME OF THE PROCEDURES AND NEED FOR PARADIGM SHIFT IN THE MANAGEMENT OF THESE CONDITIONS IN THE ELDERLY. IT IS THUS WORTH PUBLISHING

Response: Thank you for the reviewer’s positive feedback.

---

## [Decision Letter · Decision Letter 1]

6 Mar 2024

Effectiveness of percutaneous key lesion endoscopic lumbar decompression for the treatment of lumbar spinal stenosis in octogenarian patients

PONE-D-23-21132R1

Dear Dr. Cheng-Di Chiu,

We’re pleased to inform you that your manuscript has been judged scientifically suitable for publication and will be formally accepted for publication once it meets all outstanding technical requirements.

Kind regards,

Tadashi Ito

Academic Editor

PLOS ONE

Additional Editor Comments (optional):

Reviewers' comments:

Reviewer's Responses to Questions

**Comments to the Author**

1. If the authors have adequately addressed your comments raised in a previous round of review and you feel that this manuscript is now acceptable for publication, you may indicate that here to bypass the “Comments to the Author” section, enter your conflict of interest statement in the “Confidential to Editor” section, and submit your "Accept" recommendation.

Reviewer #1: All comments have been addressed

2. Is the manuscript technically sound, and do the data support the conclusions?

Reviewer #1: Yes

3. Has the statistical analysis been performed appropriately and rigorously? 

Reviewer #1: Yes

4. Have the authors made all data underlying the findings in their manuscript fully available?

Reviewer #1: Yes

5. Is the manuscript presented in an intelligible fashion and written in standard English?

Reviewer #1: Yes

6. Review Comments to the Author

Reviewer #1: The authors have taken my comments seriously. The current article seems to be suitable for publication in PLOS ONE.

7. PLOS authors have the option to publish the peer review history of their article (what does this mean?). If published, this will include your full peer review and any attached files.

Reviewer #1: No

---

## [Editor Report · Acceptance letter]

11 Mar 2024

PONE-D-23-21132R1 

PLOS ONE

Dear Dr. Chiu, 

I'm pleased to inform you that your manuscript has been deemed suitable for publication in PLOS ONE. Congratulations! Your manuscript is now being handed over to our production team.

Kind regards, 

on behalf of

Dr. Tadashi Ito 

Academic Editor

PLOS ONE